# Low-Glycemic-Index/Load Desserts Decrease Glycemic and Insulinemic Response in Patients with Type 2 Diabetes Mellitus

**DOI:** 10.3390/nu12072153

**Published:** 2020-07-20

**Authors:** Vasiliki Argiana, Panagiotis Τ. Kanellos, Ioanna Eleftheriadou, Georgios Tsitsinakis, Despoina Perrea, Nikolaos K. Tentolouris

**Affiliations:** 1Diabetes Center, First Department of Propaedeutic Internal Medicine, Medical School, National and Kapodistrian University of Athens, Laiko General Hospital, 15772 Athens, Greece; vaso.argiana@hotmail.com (V.A.); ioanna.eleftheriadou@gmail.com (I.E.); gio.tsitsi@yahoo.gr (G.T.); ntentolouris@yahoo.gr (N.K.T.); 2Laboratory of Experimental Surgery and Surgical Research N.S. Christeas, National and Kapodistrian University of Athens, 15772 Athens, Greece; dperrea@lessr.eu

**Keywords:** diabetes, glycemic index, glycemic load, glucose, insulin, xylitol

## Abstract

Diabetes mellitus is a chronic disease whose prevalence is growing worldwide. Consumption of desserts with low glycemic index (GI) and low glycemic load (GL) in a balanced hypocaloric diet has a positive impact on anthropometric and metabolic parameters in patients with type 2 diabetes mellitus (T2DM). The aim of the present study was to evaluate the glycemic and insulinemic response after consumption of desserts with low GI/GL in patients with T2DM. Fifty-one patients consumed either 83 g of the conventional dessert or 150 g of the low GI/GL dessert in random order after an overnight fast. Serum glucose, triglycerides, and insulin were measured at baseline (immediately before ingestion) and at 30, 60, 90, and 120 min postprandially. Subjective appetite measurements were performed using visual analog scales (VASs). There were significant differences at 30 (*p* = 0.014), 60 (*p* < 0.001), and 90 min (*p* < 0.001) postprandially between the two desserts for glucose and at 30 (*p* = 0.014) and 60 min (*p* = 0.033) postprandially for insulin. Glucose iAUC was significantly lower in low-GI/GL dessert compared to control (*p* < 0.001). Serum triglycerides and insulin iAUC did not differ between the two trials. Fullness VAS ratings were significantly higher after consumption of the low-GI/GL dessert compared to conventional dessert. Likewise, hunger, additional food, and additional food quantity VAS ratings were significantly lower after the consumption of the low-GI/GL dessert compared to control. Consumption of low-GI/GL dessert indicates a positive impact on metabolic parameters in T2DM patients.

## 1. Introduction

Diabetes mellitus is a chronic disease whose prevalence is growing worldwide. Some 425 million people worldwide, or 8.8% of adults aged 20–79 years, are estimated to have diabetes. In high-income countries, approximately 87% to 91% of all people with diabetes are estimated to have type 2 diabetes mellitus (T2DM). By 2045, if this trend continues, 693 million people aged 18–99 years, or 629 million people aged 20–79 years, will have diabetes [1]. The primary symptom of diabetes mellitus is elevated blood glucose levels; thus, tight glycemic control has been shown to prevent and delay associated acute and long-term complications [2]. Lifestyle management is a fundamental aspect of diabetes care and includes diabetes self-management education, diabetes self-management support, nutrition therapy, physical activity, smoking cessation counseling, and psychosocial care [3].

A recent meta-analysis revealed that a diet with a low glycemic index (GI) was found to be more effective in controlling glycated hemoglobin and fasting blood glucose compared with a higher GI diet or control in patients with T2DM [4]. In addition, a recent study conducted in the Mediterranean population showed that dietary GI and glycemic load (GL) have a potential role in the development of metabolic syndrome and associated clinical features, suggesting a promising strategy against the increasing epidemic of T2DM [5]. The GI is a quantitative measure of dietary carbohydrate quality based on the blood glucose response after consumption, in comparison with glucose [6]. A previous study has indicated that consumption of desserts with low GI and low GL in a balanced hypocaloric diet has a positive impact on anthropometric and metabolic parameters in patients with T2DM [7]. The aim of the present study was to evaluate the glycemic and insulinemic response after consumption of desserts with low GI/GL in patients with T2DM.

## 2. Materials and Methods

### 2.1. Study Population

All procedures were compliant with the Declaration of Helsinki, and the experimental protocol was approved by the local hospital ethics committee. Fifty-one men and postmenopausal women with T2DM randomly selected from the diabetes outpatient clinic of Laiko University Hospital (Athens Medical School) were recruited to the study. Recruitment was based on the following inclusion criteria: age 40 to 65 years, a minimum interval of 3 years since diagnosis, and good glycemic control (HbA1c < 7%). Diagnosis of diabetes was based on the American Diabetes Association criteria [8]. After a full explanation of the study protocol, all patients provided written consent and completed appropriate privacy authorization. All patients were under oral treatment for diabetes (metformin, sulfonylurea, gliptin). Characteristics of the study participants are displayed in Table 1.

### 2.2. Study Protocol

During the screening visit, height was measured using a stadiometer (Seca Mode 220, Germany), weight was determined in the fasting state early in the morning in light clothing using a balance scale (TanitaWB-110MA, Japan), and body mass index (BMI) was calculated by using the Quetelet index. Using the randomized crossover design, the conventional dessert and the low-GI/GL dessert were fed in random order on separate occasions with a minimum of one week between each visit in order to avoid carryover effects. Subjects were also asked to refrain from vigorous exercise the previous day of each visit. Prior to the visit, subjects had fasted overnight for 10–12 h. On arrival at the clinical investigation unit, an intravenous cannula was inserted into a forearm vein, and a baseline blood sample was taken. The subjects then consumed either 83 g of the conventional dessert or 150 g of the low-GI/GL dessert. The conventional dessert was a plain cake with sugar added while the low-GI/GL dessert was the same cake with xylitol as a sweetener instead of sugar. The amount of the available carbohydrates was calculated to be the same in both desserts tested. Low-GI/GL dessert had a GI value equal to 42 and a GL value equal to 8.8. The nutritional value of the two desserts can be seen in Table 2.

### 2.3. Blood Measurements

Venous blood samples were collected at baseline (immediately before ingestion of the dessert) and at 30, 60, 90, and 120 min postprandially. Approximately 5 mL of whole blood were obtained every time and collected into serum separator tubes. Serum glucose and triglycerides were measured immediately on an automatic analyzer. A human ELISA kit with a sensitivity range of 0.01–129 μIU/mL was used to measure serum insulin (Accubind, Los Angeles, CA, USA), and the mean intra-assay coefficient of variability was <7.0. The positive incremental area under the curve (iAUC) for serum glucose and insulin was calculated by integration. Any area beneath the fasting values was ignored [9].

### 2.4. Subjective Appetite Measurements

Visual analog scales (VASs) were used to assess subjective appetite sensation (fullness, hunger, additional food, additional food quantity, and total preference) in response to the two desserts. VASs are 10 cm in length with words anchored at each end, expressing the most positive and the most negative ratings [10].

### 2.5. Statistical Analysis

The Statistical Package for the Social Sciences (SPSS 25.0 for Windows, Chicago, IL, USA) was used for all the analyses. For all measures, descriptive statistics were calculated. Results are expressed as means ± standard error of the mean (SEM). All variables were tested for normal distribution of the data applying the Kolmogorov–Smirnov test. Differences among treatments, among time points, and the interaction between treatment and time were tested by paired *t*-test (2-sided). The level of significance was set at *p* < 0.05.

## 3. Results

There was no significant difference in baseline serum glucose levels between the two trials. Serum glucose peaked significantly at 90 min for both low-GI/GL and conventional dessert (Figure 1a). There were significant differences at 30 (155.3 ± 4.4 vs. 164.2 ± 4.2 mg/dL, *p* = 0.014), 60 (172.9 ± 4.6 vs. 195.6 ± 5.9 mg/dL, *p* < 0.001) and 90 min (184 ± 5.1 vs. 202.6 ± 6.3 mg/dL, *p* < 0.001) postprandially between the low-GI/GL dessert and the conventional one. Likewise, serum insulin peaked at 90 min for both desserts (Figure 1b). There was no significant difference in baseline serum insulin levels between the two trials. There were significant differences at 30 (21.8 ± 3.2 vs. 26.4 ± 3.2 μIU/mL, *p* = 0.014) and 60 min (30 ± 2.9 vs. 35.9 ± 3.4 μIU/mL, *p* = 0.033) postprandially between the two desserts. Serum triglycerides did not indicate any difference between the two trials (Figure 1c). Glucose iAUC (Figure 2a) was significantly lower (*p* < 0.001) after low-GI/GL dessert consumption compared to control; no difference was observed in insulin iAUC (Figure 2b).

Fullness VAS ratings were significantly (*p* = 0.004) higher after consumption of the low-GI/GL dessert (7.5 ± 0.3 cm) compared to conventional dessert (6.3 ± 0.4 cm). Likewise, hunger, additional food, and additional food quantity VAS ratings were significantly lower (*p* < 0.05) after ingestion of the low-GI/GL dessert compared to control (1.6 ± 0.2 vs. 2.7 ± 0.4 cm, 2.5 ± 0.3 vs. 3.6 ± 0.4 cm, and 2.7 ± 0.3 vs. 3.3 ± 0.3 cm, respectively) (Table 3).

## 4. Discussion

Our results indicate that consumption of a dessert with a low glycemic index/glycemic load ameliorates glucose and insulin responses in patients with type 2 diabetes compared to a conventional dessert with similar content of available carbohydrates but different sugar and fiber content.

Xylitol is the main sweetener of the low-GI/GL dessert, replacing the sugar contained in the conventional dessert. Xylitol belongs to the group of sugar alcohols, polyols, in which the carbonyl moiety (- – - C = O) of carbohydrates is replaced by an alcohol radical (- – - CH-OH) [11]. It is present in a very small quantity in fruits, such as plums, strawberries, and raspberries, and vegetables, such as cauliflower, pumpkin, and spinach, and it is produced commercially by hydrogenation of xylose in a nickel-catalyzed process. Due to the fact that this process is difficult, costly, and energy intensive, some alternative biotechnological processes have been studied, involving yeasts from *Candida* genus [12]. Early enough after consumption, literature has indicated that xylitol can be metabolized independently of insulin because it is not transported actively through the intestinal tract and does not require insulin for uptake by the liver; therefore, it gives a low glycemic index. After xylitol is absorbed into the bloodstream, liver uptake of xylitol is insulin-independent and causes very little increase in blood glucose, insulin, and glucagon levels [12]. It was initially considered that xylitol’s lack of insulin responses would not be apparent when eaten with a complex meal.

Herein, consumption of a dessert with xylitol had a favorable impact on glycemic and insulinemic responses compared to a dessert with sucrose. In accordance with our findings, in a study where 30 g of xylitol or 30 g of sucrose was used to substitute starch in a standardized meal as a part of diabetic diet, xylitol led to lower blood glucose and insulin responses compared to sucrose in the context of a more complex meal [13]. In a very recent study, 10 lean and 10 obese volunteers were given 75 g of glucose and 50 g of xylitol in 300 mL of water or placebo (water) by a nasogastric tube. Xylitol intake did not affect glucose response in lean subjects, while it significantly increased plasma glucose response (AUC_0–180 min_) in obese individuals, suggesting that obesity is the effect modifier. Regarding insulinemic responses, xylitol had a minimal but statistically significant enhancing effect on insulin AUC_0–180 min_ in both lean and obese individuals (*p* < 0.001 and *p* = 0.047, respectively) [14].

Another difference between the two desserts tested was the dietary fiber content. More specifically, low-GI/GL dessert contains 8 g of dietary fiber per 100 g of dessert. In comparison, the conventional dessert contains to 0.7 g of dietary fiber per 100 g of dessert. In 1972, Trowell suggested that dietary fiber consists of the remnants of edible plant cells, polysaccharides, lignin, and associated substances resistant to digestion by the alimentary enzymes of humans [15]. Dietary fiber is resistant to digestion and absorption in the human small intestine and comes in two main forms depending on its solubility in water (soluble and insoluble fiber) [16]. Dietary fiber intake is considered to improve glycemic control, lower plasma lipid concentrations, and reduce postprandial insulinemia in patients with type 2 diabetes [17,18]. Traditional societies consumed largely unprocessed plant-based diets that were high in fiber and included whole grains, legumes, and nuts as staples; these diets had low GI and GL. Epidemiological studies indicate that the consumption of plant-based diets reduces the risk of diabetes and cardiovascular diseases [19]. The “fiber hypothesis” suggested that this is a direct effect of fiber [20]. The GI concept is an extension of the fiber hypothesis, suggesting that fiber reduces the rate of nutrient influx from the gut [21].

A meta-analysis that included data from 11 intervention studies lasting four weeks or longer concluded that low-GI diets improved glycemic control and body peripheral insulin sensitivity and reduced HbA1c to an extent that was comparable to medication in newly diagnosed T2DM patients [22]. Silva et al. tested four different breakfasts in 14 patients with type 2 diabetes (high GI/high fiber, high GI/low fiber, low GI/high fiber, and low GI/low fiber). Plasma glucose, insulin, and ghrelin responses were least favorable when patients consumed a breakfast with high GI and low fiber, which suggests that reducing the GI of breakfasts, increasing the fiber content of breakfasts, or both may be useful strategies for improving the postprandial metabolic profile of these patients [23].

One potential point of criticism for the GI concept is that the glycemic impact of foods may be affected by certain meal components. It has been shown that adding whey and/or certain amino acids to a meal may substantially reduce the glycemic response in healthy and T2DM subjects [24]. However, for foods and meals rich in carbohydrates, the effects of additive components of GI are minor, supporting the utility of the GI concept also for mixed meals in T2DM subjects [25].

Moreover, in our study, visual analog scales were used to assess subjective appetite sensation in response to the two desserts. According to these ratings, the low-GI/GL dessert induced greater subjective fullness and less hunger, additional food, and additional food quantity than the control. Glucose and especially insulin response have been associated with appetite, hunger, and/or satiety [26]. After a meal, the highest postprandial glucose increment and its earliest and sharpest decline seem to be key for the onset of hunger [27]. Therefore, the lower glucose and insulin peaks after low-GI/GL dessert could explain the differences in VAS ratings. In Figure 1a,b, it is also obvious that the sharpest decline in glucose and insulin levels occurs at 180 min after ingestion of the conventional dessert, boosting the hunger sensation. In the study of Wölnerhanssen et al., xylitol intake led to a significant increase in glucagon-like peptide-1 and cholecystokinin, suggesting a stimulation of gut hormone release and decrease in gastric emptying; however, subjective feelings of appetite did not differ significantly compared to the placebo [14]. Similarly, in the study of Silva et al., despite changes in postprandial glucose and insulin, no effect on subjective measures of hunger was observed [23].

## 5. Conclusions

A dessert with low glycemic index and low glycemic load containing xylitol as sweetener and high fiber content seems to be favorable in terms of glucose and insulin responses when compared to a conventional dessert that contains sugar and no fiber. In combination with its good flavor and the fact that it seems to suppress hunger and induce satiety, such a dessert could be a good choice for patients with type 2 diabetes or people who are trying to lose weight or maintain an ideal weight.

## Figures and Tables

**Figure 1 nutrients-12-02153-f001:**
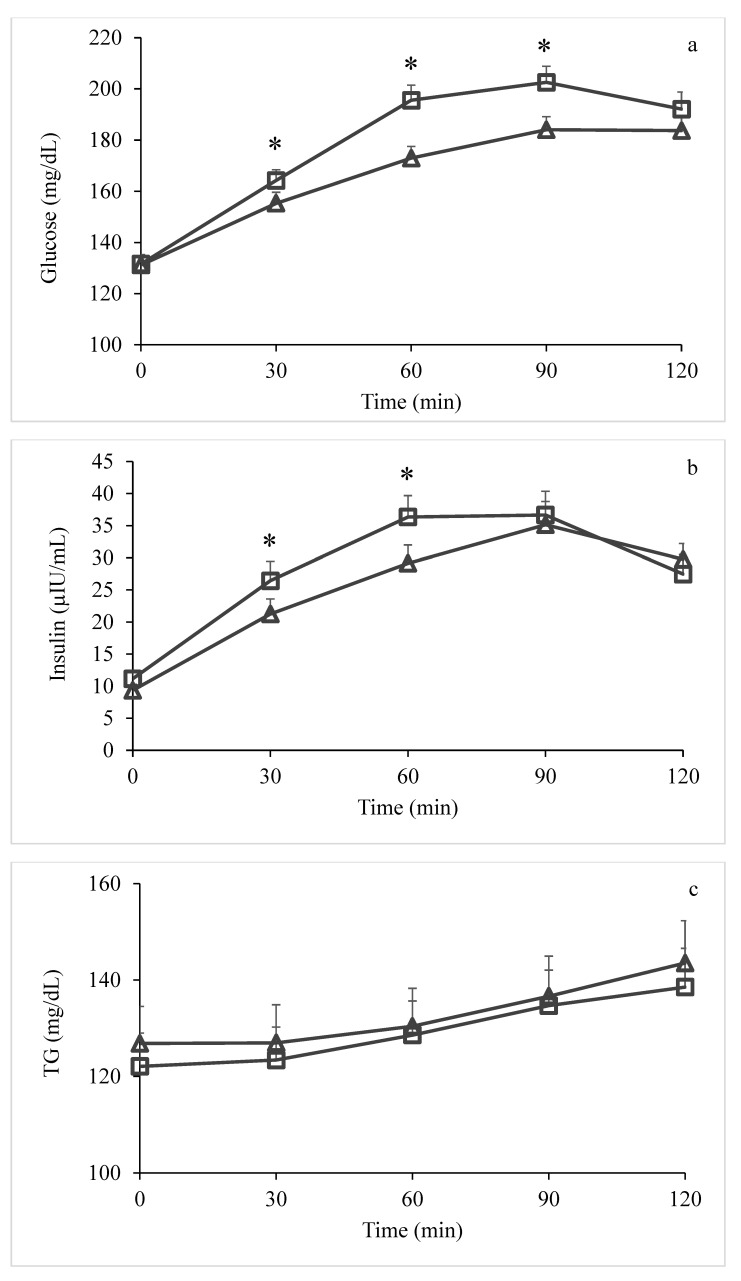
Serum glucose (**a**), serum insulin (**b**), and serum triglycerides (**c**) (mean ± SEM) following consumption of low-GI/GL dessert (Δ) and conventional dessert (□). Statistical significance (*p* < 0.05) between response to low-GI/GL dessert and control is indicated by *.

**Figure 2 nutrients-12-02153-f002:**
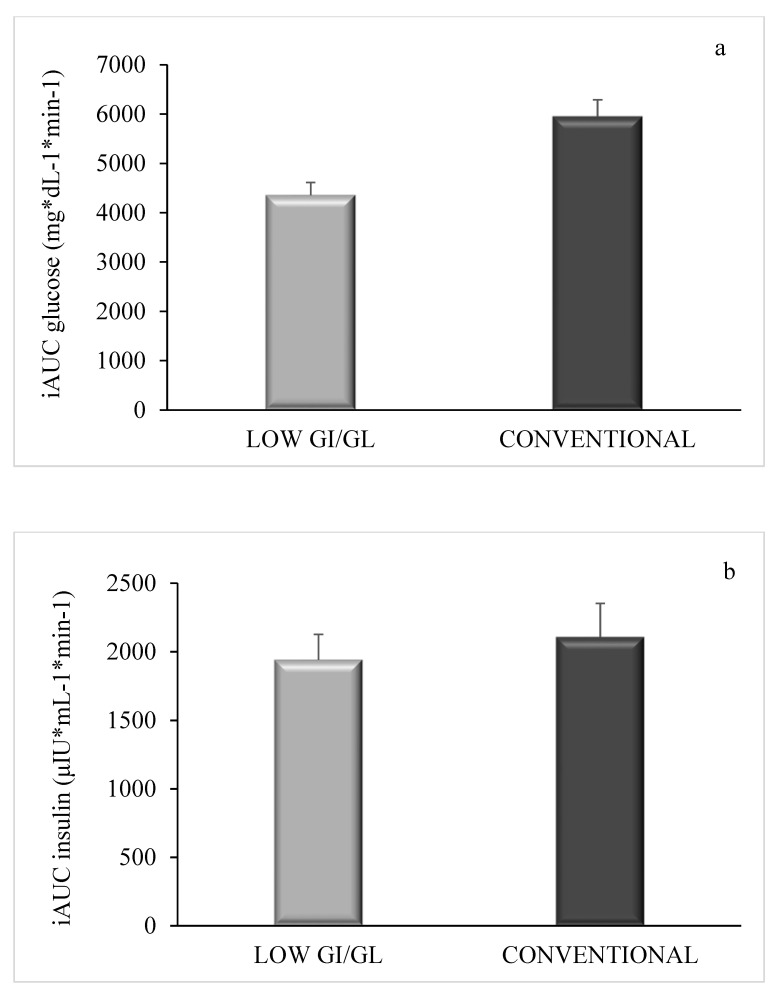
Incremental area under the curve (iAUC) for glucose (**a**) and insulin (**b**) (mean ± SEM) following consumption of low-GI/GL and conventional desserts. Statistical significance (*p* < 0.05) between response to low-GI/GL dessert and control is indicated by *.

**Table 1 nutrients-12-02153-t001:** Baseline characteristics of the study participants.

*N*	51
Gender, *n* Males Females	23 28
Age, year	62 ± 1
Smoking, *n* (%)	10 (19.6)
Body weight, kg	84.8 ± 1.9
BMI, kg/m^2^	31.6 ± 0.6
Waist circumference, cm	104.4 ± 1.4
Hip circumference, cm	109.8 ± 1.1
SBP, mmHg	131.5 ± 1.9
DBP, mmHg	75.9 ± 1.1
HbA1c, %	6.4 ± 0.1
Total chol., mg/dL	171.7 ± 3.6
LDL chol., mg/dL	98.8 ± 4.4
HDL chol., mg/dL	45.3 ± 1.2
TG, mg/dL	126.8 ± 7.7
SGOT, U/L	22.2 ± 0.9
SGPT, U/L	22 ± 1.2
γ-GT, U/L	23.4 ± 1.6

Data are mean values ± standard error of mean (SEM) or n (%); BMI: body mass index; HbA1c: glycated hemoglobin; SBP: systolic blood pressure; DBP: diastolic blood pressure; chol: cholesterol; LDL: low-density lipoprotein; HDL: high-density lipoprotein; TG: triglycerides; SGOT: serum glutamic oxaloacetic transaminase; SGPT: serum glutamic pyruvic transaminase; γ-GT: gamma-glutamyl transpeptidase.

**Table 2 nutrients-12-02153-t002:** Nutritional value per 100 g of the two desserts (conventional and low glycemic index/load) consumed by patients with type 2 diabetes.

Nutrient	Low-GI/GL Dessert	Conventional Dessert
Energy (kcal/kJ)	332/1389	407/1711
Protein (g)	7.0	6.5
Carbohydrates (g) Sugars (g)	42.5 0.3	59.2 33.4
Polyols (g)	14.3	-
Total fat (g) Saturated (g)	17.1 3.8	15.9 7.2
Dietary fibers (g)	7.9	0.7
Sodium (g)	0.27	0.26

**Table 3 nutrients-12-02153-t003:** Ratings for fullness, hunger, additional food, additional food quantity, and total preference after consumption of the two desserts.

	Low-GI/GL Dessert	Conventional Dessert	*p*-Value
Fullness (cm)	7.5 ± 0.3	6.3 ± 0.4	0.004
Hunger (cm)	1.6 ± 0.2	2.7 ± 0.4	0.001
Additional food (cm)	2.5 ± 0.3	3.6 ± 0.4	0.002
Additional food quantity (cm)	2.7 ± 0.3	3.3 ± 0.3	0.016
Preference (cm)	6.5 ± 0.4	6.9 ± 0.4	0.275

Data are mean values ± SEM; *p*-value: comparison between the two desserts by the paired samples *t*-test.

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
