# Peer review of "Low-Glycemic-Index/Load Desserts Decrease Glycemic and Insulinemic Response in Patients with Type 2 Diabetes Mellitus"

_nutrients, 2020, doi:10.3390/nu12072153_

Round 1
Reviewer 1 Report
Tentolouris and team in their report studied the impact of low glycemic load containing dessert vs high glycemic load containing dessert in a cohort of T2D subjects. The study involves sufficient number of subjects to make a meaningful conclusion. The study involves both male as well as female subjects.
One of the key issue that hold the manuscript back is the mean age of the subjects (62 years). Why younger subjects were not included in the study should be mentioned in the discussion.
The data is robust however, results are not surprising. Fiber has been know to influence glucose absorption and insulin response. Authors report suggest fiber and xylitol are the two important factors that results in improved insulin sensitivity and glucose tolerance in the subjects.
- Does authors have any previous record of the diet that subjects consumed? Was the diet rich in fiber?
- The term usual dessert should be replaced with more scientific terminology? What is the definition of the usual dissert? Is it specific to a particular region?
- Is there any correlation between smoking and LI vs usual diet found in this study?
The data is clean and make sense. The discussion needs some elaboration as mentioned above. The study is well within the scope of the journal and may be accepted after minor revision. I congratulate authors for the good job.
Author Response
Reviewer(s)' Comments to Author:
Replies to Reviewer(s)’ queries
Tentolouris and team in their report studied the impact of low glycemic load containing dessert vs high glycemic load containing dessert in a cohort of T2D subjects. The study involves sufficient number of subjects to make a meaningful conclusion. The study involves both male as well as female subjects.
One of the key issue that hold the manuscript back is the mean age of the subjects (62 years). Why younger subjects were not included in the study should be mentioned in the discussion.
The authors wish to thank Reviewer for the question. The sample used in our study was randomly selected from the diabetes outpatient clinic of Laiko University Hospital. There is no other reason for the age of the patients which enrolled to the study. If Reviewer wishes to add a comment to the text authors could follow his suggestion.
The data is robust however, results are not surprising. Fiber has been know to influence glucose absorption and insulin response. Authors report suggest fiber and xylitol are the two important factors that results in improved insulin sensitivity and glucose tolerance in the subjects.
- Does authors have any previous record of the diet that subjects consumed? Was the diet rich in fiber?
The authors wish to thank Reviewer for the comment. Unfortunately, authors have no data about the diet of patients which they followed prior to the study.
- The term usual dessert should be replaced with more scientific terminology? What is the definition of the usual dissert? Is it specific to a particular region?
The authors wish to thank Reviewer for the comment. As it is referred in the text the dessert was a plain cake either “usual” or “with low GI/GL. The nutritional value of the two desserts is presented in Table 2. The term “usual” is now replaced by the term “conventional” according to Reviewer’s comment.
- Is there any correlation between smoking and LI vs usual diet found in this study?
The authors wish to thank Reviewer for the comment. According to our analysis no differences were reported as far as smokers and non-smoker patients who participated in our study.
The data is clean and make sense. The discussion needs some elaboration as mentioned above. The study is well within the scope of the journal and may be accepted after minor revision. I congratulate authors for the good job.
The authors wish to thank Reviewer for the comments. All amendments have now been done according to the Reviewer’s suggestion.

Reviewer 2 Report
Rates of obesity continue to increase across the western hemisphere, and to a lesser extent in the east. Obesity causes a host of diseases including type 2 diabetes. By 2050, roughly 30% of Americans are projected to develop type 2 diabetes. As calorie-rich dietary patterns are unlikely to drastically change, novel nutrition therapies are needed that can help control dysglycemia and hyperinsulinemia. The authors basically used an OGTT approach to gauge baseline and post-prandial glucose, triglyceride, and insulin, except that changes were based on consuming a standard and low glycemic dessert, with participants acting as their own controls after an overnight fast. The authors found that the low glycemic desert not only reduced glucose and insulin levels, but led to less appetitive drive and feelings of fullness.
In general, this straightforward, highly useful study may help to address how people can negotiate consuming dessert, which is a common component of the western diet. I have one minor suggestion that is optional. Otherwise, the report speaks for itself.
Abstract
*This is an excellent summary of the data. I have no suggested changes.
Introduction
*While the increased number of people who have type 2 diabetes in 2045 relative to now is accurate, percentages may be more intuitive and would correct for increases in world population.
*The background is short and concise, which is always appreciated, and presents a good rationale for conducting the current study, to fill the gap of how people with type 2 diabetes respond to a standard versus low glycemic index/load dessert. While there is some response to the low GI/GL dessert despite having the insulin-independent sugar, this is to be expected because any complex meal will result in increased output by the pancreas to process any nutrients that are insulin-reliant for processing and uptake.
Methods
*While it would be interesting to see how a low glycemic index/load dessert might affect people with poorly controlled type 2 diabetes or more recent diagnosis, it’s important to first examine people with type 2 diabetes who have “optimal” control to guard against much wider differences in post-prandial measurements that could obscure results.
*The crossover design implemented here is easy to do and allows for much smaller sample sizes while maintaining comparable Power.
*The protocol itself is ideal for avoiding confounding effects of exercise, variation in circadian rhythms that may be relevant, and other related concerns.
*The desserts appear to be well balanced and involves only a difference in the sweetener used (sugar vs. xylitol). The much higher dietary fiber would at least in part explain increased feelings of post-prandial fullness.
*The insulin kit has a decent CV and likely would not have ceiling or basement effects.
*The statistics are straightforward. If a more subtle effect was expected, I might have suggested a mixed linear regression model to take time into account, but the signal here is obvious and thus this first step is not necessary.
Results
*The results for glucose and insulin are about as expected, within glucose being much lower for low GI/GL dessert, and insulin being significant for a shorter period of time simply due to the nature of digestion for this type of food item. This obviously would explain the lack of significance for the insulin AUC. The differences are clear and easily interpretable.
*Likewise, Table 3 makes it very clear that there are visual ratings differences between the two desserts.
*A benefit to having a crossover study is that variation in glucose or insulin cannot be attributed to individual differences in processing and absorption.
Discussion
*The remarks about xylitol and why one sees relatively lower secretion of insulin are accurate and this literature summarized well.
*The rest of Discussion does a fine job discussing dietary fiber, relevant meta-analyses involving high vs. low fiber and low vs. high GI foods. While some readers or reviewers might question GI and what it represents, I consider this fairly well established science. From the most simplistic point of view: if one needs less insulin secretion to process a food, insulin resistance will not proceed as quickly. Thus, a unit of insulin is more likely to facilitate adequate amounts of glucose uptake into somatic muscle and certain GLUT4-mediated areas of the brain to maintain function.
*The use of VAS is fine. Anyone suggesting that objective measurement of fullness are needed, such as an index of gastric emptying or other complicated parameters related to appetitive drive, is missing the point of the report. Taken to its extreme, one could argue for assaying leptin, ghrelin, and hypothalamic releasing or acting hormones to establish if the body is signaling that it is full. This is not at all feasible, practical, or important. In fact, I would argue that VAS is superior to objective measures of fullness because the person lacks the appetitive drive to pursue more food or wait a longer span of time to eat again, regardless of how objectively satiated that person is.
*The rest of Discussion and the Conclusion are all solidly grounded in the findings.
Author Response
Reviewer(s)' Comments to Author:
Replies to Reviewer(s)’ queries
Rates of obesity continue to increase across the western hemisphere, and to a lesser extent in the east. Obesity causes a host of diseases including type 2 diabetes. By 2050, roughly 30% of Americans are projected to develop type 2 diabetes. As calorie-rich dietary patterns are unlikely to drastically change, novel nutrition therapies are needed that can help control dysglycemia and hyperinsulinemia. The authors basically used an OGTT approach to gauge baseline and post-prandial glucose, triglyceride, and insulin, except that changes were based on consuming a standard and low glycemic dessert, with participants acting as their own controls after an overnight fast. The authors found that the low glycemic desert not only reduced glucose and insulin levels, but led to less appetitive drive and feelings of fullness.
In general, this straightforward, highly useful study may help to address how people can negotiate consuming dessert, which is a common component of the western diet. I have one minor suggestion that is optional. Otherwise, the report speaks for itself.
The authors wish to thank Reviewer for the nice comments.
Abstract
*This is an excellent summary of the data. I have no suggested changes.
Introduction
*While the increased number of people who have type 2 diabetes in 2045 relative to now is accurate, percentages may be more intuitive and would correct for increases in world population.
*The background is short and concise, which is always appreciated, and presents a good rationale for conducting the current study, to fill the gap of how people with type 2 diabetes respond to a standard versus low glycemic index/load dessert. While there is some response to the low GI/GL dessert despite having the insulin-independent sugar, this is to be expected because any complex meal will result in increased output by the pancreas to process any nutrients that are insulin-reliant for processing and uptake.
Methods
*While it would be interesting to see how a low glycemic index/load dessert might affect people with poorly controlled type 2 diabetes or more recent diagnosis, it’s important to first examine people with type 2 diabetes who have “optimal” control to guard against much wider differences in post-prandial measurements that could obscure results.
*The crossover design implemented here is easy to do and allows for much smaller sample sizes while maintaining comparable Power.
*The protocol itself is ideal for avoiding confounding effects of exercise, variation in circadian rhythms that may be relevant, and other related concerns.
*The desserts appear to be well balanced and involves only a difference in the sweetener used (sugar vs. xylitol). The much higher dietary fiber would at least in part explain increased feelings of post-prandial fullness.
*The insulin kit has a decent CV and likely would not have ceiling or basement effects.
*The statistics are straightforward. If a more subtle effect was expected, I might have suggested a mixed linear regression model to take time into account, but the signal here is obvious and thus this first step is not necessary.
Results
*The results for glucose and insulin are about as expected, within glucose being much lower for low GI/GL dessert, and insulin being significant for a shorter period of time simply due to the nature of digestion for this type of food item. This obviously would explain the lack of significance for the insulin AUC. The differences are clear and easily interpretable.
*Likewise, Table 3 makes it very clear that there are visual ratings differences between the two desserts.
*A benefit to having a crossover study is that variation in glucose or insulin cannot be attributed to individual differences in processing and absorption.
Discussion
*The remarks about xylitol and why one sees relatively lower secretion of insulin are accurate and this literature summarized well.
*The rest of Discussion does a fine job discussing dietary fiber, relevant meta-analyses involving high vs. low fiber and low vs. high GI foods. While some readers or reviewers might question GI and what it represents, I consider this fairly well established science. From the most simplistic point of view: if one needs less insulin secretion to process a food, insulin resistance will not proceed as quickly. Thus, a unit of insulin is more likely to facilitate adequate amounts of glucose uptake into somatic muscle and certain GLUT4-mediated areas of the brain to maintain function.
*The use of VAS is fine. Anyone suggesting that objective measurement of fullness are needed, such as an index of gastric emptying or other complicated parameters related to appetitive drive, is missing the point of the report. Taken to its extreme, one could argue for assaying leptin, ghrelin, and hypothalamic releasing or acting hormones to establish if the body is signaling that it is full. This is not at all feasible, practical, or important. In fact, I would argue that VAS is superior to objective measures of fullness because the person lacks the appetitive drive to pursue more food or wait a longer span of time to eat again, regardless of how objectively satiated that person is.
*The rest of Discussion and the Conclusion are all solidly grounded in the findings.
The authors appreciate Reviewer’s flattering comments and thorough analysis of all sections of our paper. As far as authors understand, Reviewer does not suggest any further changes.